# Quality Improvement Project to Improve Hand Hygiene Compliance in a Level III Neonatal Intensive Care Unit

**DOI:** 10.3390/children10091484

**Published:** 2023-08-30

**Authors:** Pavani Chitamanni, Ahreen Allana, Ivan Hand

**Affiliations:** 1Department of Pediatrics, NYC Health & Hospitals/Kings County, Brooklyn, NY 11203, USA; pavani.chitamanni@downstate.edu (P.C.); ahreen_allana@hotmail.com (A.A.); 2Department of Pediatrics, SUNY-Downstate Health Sciences University, Brooklyn, NY 11203, USA

**Keywords:** NICU, hand hygiene, PDSA cycle, healthcare-associated infections

## Abstract

This quality improvement project aimed to improve hand hygiene (HH) compliance in a Level III Neonatal Intensive Care Unit. The project was conducted over three Plan–Do–Study–Act (PDSA) cycles, with each cycle lasting two months. The interventions included healthcare worker (HCW) education on HH, repetition of education, and immediate feedback to HCWs. Compliance data were collected through covert observations of HCWs in the NICU. The overall compliance rate increased from 31.56% at baseline to 46.64% after the third PDSA cycle. The HH compliance was noted to be relatively low after touching patient care surroundings, at entry and exit from the NICU main unit, before wearing gloves and after removing gloves, at baseline and throughout the three PDSA cycles. HCW education alone did not result in significant improvements, highlighting the need for additional interventions. The study underscores the importance of involving NICU leadership and providing immediate feedback to promote HH compliance. Further efforts should focus on addressing the false sense of security associated with glove usage among HCWs, individual rewards and involving the healthcare staff in the shared goal of increasing HH compliance. Consideration of workload metrics and their impact on compliance could steer future interventions.

## 1. Introduction

Healthcare-associated infections (HAIs) are one of the most common complications of healthcare. They were initially accepted as inevitable complications of healthcare and hospitalization. However, HAIs were later understood to be preventable, and there has been a significant improvement in interventions aimed at decreasing their incidence. According to recent CDC (Centers for Disease Control and Prevention) data, HAIs still occur in one out of 31 hospitalized patients every day. The economic impact of HAIs is huge with an estimated cost of billions of dollars annually [1].

Hand hygiene (HH) is a simple and effective measure to decrease the spread of hospital-acquired infections. Investigators have demonstrated reductions in HAI and MDRO (Multidrug-Resistant Organisms) infections when compliance increased from low to medium levels (48% to 66%) [2]. Improving adherence to hand hygiene in a healthcare facility is of critical importance, especially in a neonatal intensive care unit (NICU), where premature immunocompromised neonates receive care. Given the importance of this simple and affordable measure, it should be constantly promoted. HH has gained more importance in the COVID era. Although the HH adherence rates initially increased as the pandemic began, they later decreased as the pandemic progressed [3].

Previous studies show that the importance of HH has been underestimated. There exists a knowledge gap on when and how to perform hand hygiene among healthcare providers, irrespective of their understanding of its importance, while others have misconceptions about the use of hand sanitizer [4]. The World Health Organization (WHO) promoted the “Five Moments” model on when to perform hand hygiene in the healthcare setting [5]. The five moments include: before touching a patient, before a clean/aseptic procedure, after body fluid exposure risk, after touching a patient, and after touching patient surroundings. Despite the knowledge of the importance and timing of hand hygiene, 100% adherence may not be attainable. The different reasons for poor hand hygiene identified include lack of time, high workload, shortage of nurses, inadequate time management, skin irritation, unavailability of materials in the work area or in appropriate places, lack of training and experience, lack of feedback, lack of role models in the team, lack of performance rewards or sanctions, lack of motivation at individual or institutional level and lack of continuous training [6].

There were many studies and Quality Improvement projects, which looked at interventions to bring about a change in the hand hygiene behavior among healthcare providers. The various interventions include training and education, observation and feedback, reminders in the hospital, etc. [6]. The compliance rates increased when overt observations were made as opposed to covert observations as per a study conducted in Israel [7]. However, such an intervention might not be sustainable, given overt observations cannot be made all the time. On the other hand, a system change can integrate hand hygiene behavior into healthcare. For example, a study conducted at Stanford Hospital demonstrated that observing attending physicians with good hand hygiene practices was the most effective strategy in influencing trainees including residents and medical students [8]. This QI project was aimed at bringing about a consistent behavior change toward hand hygiene. The interventions were conducted in a stepwise manner so that the impact of each intervention can be assessed.

## 2. Materials and Methods

This Quality Improvement Initiative was reviewed and determined to not meet the criteria for human subjects research by the SUNY Downstate Institutional Review Board. The goal of our QI project was to improve adherence to hand hygiene in the NICU at NYC Health + Hospitals, Kings County, by 20% within a six-month period. We conducted this Quality Improvement (QI) project in the 30-bedded Level III NICU at NYC Health + Hospitals, Kings County. We used the Plan–Do–Study–Act (PDSA) problem-solving approach. We implemented 3 PDSA cycles, with each PDSA cycle implemented over two months.

An interdisciplinary team was created to assess HH compliance in the unit. The investigators who participated in recording the observations have completed the hand hygiene observer and coach training modules, which are available in the ‘My Learning’ section of the NYCHHC Employee Self Service platform. To evaluate interobserver variability, HH compliance among a group of healthcare workers (HCWs) in the NICU was assessed by multiple observers on the same day. The findings revealed that the level of variability between the observers was less than 5%. Nine HH opportunities were observed: (a) before touching a patient without gloves; (b) after touching a patient without gloves; (c) before a clean/aseptic procedure; (d) after body fluid exposure; (e) after touching the patient surroundings; (f) at entry into the NICU main unit; (g) at exit from the NICU main unit; (h) before wearing gloves; and (i) after removing gloves. The patient surroundings monitored were radiant warmer or isolette, mechanical ventilator, infusion pumps or lines and the patient monitor of the patient. 

Data were collected by direct covert observations of the HCWs in the NICU at NYC Health + Hospitals, Kings County, including attending physicians, resident physicians in various levels of training, neonatology fellows and nurses. No more than 3 observations of the same employee were made in a single observation session. Emergent/urgent situations were exempt from hand hygiene observations. When an observation was recorded, along with the specific opportunity for HH, data were also collected on the time of the observation (day/night) and the role of the healthcare provider including attending physician, resident physician and registered nurse. The level of training of the residents being observed was also noted. The healthcare workers’ names were not documented in any records. If the HCW was non-compliant with HH, the obvious reasons for non-compliance were noted, including an inconveniently located dispenser, empty or broken dispenser, improper use of gloves, distracted HCW, HCWs rounding in a group, multiple entries and exits, carrying equipment or medications into the unit. These anonymous observations were recorded on an Excel sheet and also in the New York City Health and Hospitals Corporation (NYCHHC) hand hygiene database. Access to this data was limited only to the investigators and other NYCHHC hand hygiene observers and coaches. Data collected during this QI project was kept confidential and used for QI project purposes only. 

### 2.1. Interventions

The interventions were implemented stepwise so that the relative effect of each intervention can be assessed. 

#### 2.1.1. PDSA Cycle 1 and First Intervention

Our first intervention was hand hygiene education for the healthcare workers (HCW) in our unit. We used the “five moments” model for HH by the WHO to educate the HCWs on the various opportunities for hand hygiene. Analysis of the baseline compliance data revealed five specific areas where HH compliance scored low. They were after touching patient surroundings, at entry into the NICU main unit, at exit from the NICU main unit, before wearing gloves, and after removing gloves. These specific opportunities for improvement were emphasized during the HCW education sessions. The importance and misconceptions of HH as well as the NYCHHC glove and nail policies were also discussed. Education was carried out using presentations and flyers. The Power Point presentations were prepared based on the Centers for Disease Control and Prevention (CDC) guidelines for hand hygiene in healthcare settings, World Health Organization (WHO) and NYCHHC policies. The presentations were given to HCWs individually when they were available without interfering with the patient care. Printouts of the guidelines for hand hygiene in healthcare settings and ‘the science behind these recommendations’ by CDC were given to the HCWs when requested. The power point presentations also included the baseline compliance rates at the nine different HH opportunities in our unit. Of note, brief individual surveys were conducted among the staff to assess the knowledge, perceptions, and barriers to HH just before giving HH education. 

#### 2.1.2. PDSA Cycle 2 and Second Intervention

Our second intervention was to repeat HH education for the HCWs in our unit. Repeat education was completed using the same presentations and flyers used for education in the first PDSA cycle. Individual surveys were conducted again among the staff to assess the knowledge, perceptions, and barriers to HH before repeating HH education. 

#### 2.1.3. PDSA Cycle 3 and Third Intervention

Our third intervention was immediate feedback to the individual HCWs by the NICU head nurses. Both positive and negative feedback was provided for those who performed hand hygiene or missed an opportunity for hand hygiene as appropriate. The feedback was given in a timely, specific, and respectful manner. Immediate feedback was provided both during the day and night shifts. We designed an immediate feedback form (Figure 1) to document the number of times immediate feedback was given for each HH opportunity over the day. We aimed to provide immediate feedback on a minimum of 20 occasions per day. These feedback forms were incorporated into the daily workflow of the NICU head nurses. 

### 2.2. Key Measures

#### 2.2.1. Outcome Measures

The overall compliance with HH in the unit, which is calculated as the percentage of times HH was performed out of the total number of observed opportunities for HH in a day.Compliance rates for each of the nine HH opportunities observed. This was calculated as the percentage of times HH was performed for a specific HH opportunity out of the total number of observed opportunities for that HH opportunity in a day.

#### 2.2.2. Process Measures

PDSA 1 and 2 cycles: Number of educational interventions, including presentations and flyers given.PDSA 3 cycle: Number of times immediate feedback was given in a day.

#### 2.2.3. Balancing Measures

Perceived increase in work burden and interference with patient care by HCWs. This was assessed using individual surveys conducted among the HCWs during PDSA cycles 2 and 3.

### 2.3. Statistical Considerations

#### 2.3.1. Sample Size

We calculated the total number of observations for each PDSA cycle as 93 for power of a 80% and 95% confidence level. However, the World Health Organization (WHO) manual for HH observation recommends observing a minimum of 200 opportunities for HH during each measurement period [9]. Therefore, we planned to collect a minimum of 200 observations for each PDSA cycle. We also adopted the WHO’s recommendations for the data collection plan, where we selected random days each week to observe hand hygiene compliance, collected a minimum of 15 observations of hand hygiene opportunities on the selected days, and computed the percentage of hand hygiene compliance for that week [10].

#### 2.3.2. Analysis

We used descriptive statistics to assess the pre- and post-intervention percentage of HH compliance. In each PDSA cycle, we compared the overall compliance percentage with HH before and after intervention among various healthcare provider roles (physicians vs. residents at different levels of training vs. neonatology fellows vs. nurses). We calculated the compliance percentage for each of the nine HH opportunities before and after intervention, to assess the effect of the intervention on the specific HH opportunities. The data were also plotted each week on a run chart to interpret whether the variations were random or non-random.

## 3. Results

We had a total of 377 observations at baseline, 319 for the first PDSA cycle, 239 for the second PDSA cycle and 298 for the third PDSA cycle. The baseline overall compliance was 31.56%. After the first intervention, HH compliance increased to 37.61%, and it was 33.96% at the end of the second PDSA cycle. The compliance increased to 46.64% after the third PDSA cycle (Table 1).

From the data in Table 1, it is evident that the first and second interventions, which were HCW education, did not have a significant impact on HH compliance in our unit. Giving immediate feedback, which was the third intervention, had a significant impact on the overall HH compliance.

The data in Table 2 show that giving immediate feedback had significant impact on almost all the HH opportunities, except for after touching a patient’s body fluids and before wearing gloves. The *p*-value for increase in HH compliance after touching a patient without gloves and after touching a patient surroundings is borderline. This could be due to the low power of the study for each HH opportunity.

We compared the overall compliance percentage with HH before and after intervention, among various healthcare provider roles including physicians, residents at different levels of training, neonatology fellows and nurses (Table 3 and Table 4). Giving immediate feedback for HH had a positive impact on all the HCWs.

When the compliance data were plotted on a run chart every week (Figure 2), six data points were above the median compliance line after implementing the third intervention. This is suggestive of a non-random variation due to the third intervention, which was giving immediate feedback individually to the HCWs at various opportunities for HH.

The results of the individual surveys conducted at baseline showed that the median self-reported compliance by HCWs was >80%. About 59% of the HCWs believed that soap and water is the recommended method of HH by the CDC in healthcare settings. More than 90% of the HCWs correctly identified all the opportunities for hand hygiene on a multiple-choice question on the survey. Common barriers to HH identified were increased work burden (83.3%), skin dryness and irritation (66.7%) and empty or broken sanitizer dispensers (66.7%).

## 4. Discussion

Hand hygiene (HH) is a simple yet extremely important measure to decrease healthcare-associated infections, more so in a Neonatal Intensive Care Unit. Our QI project aimed at improving HH compliance in a level 3 NICU. Although the WHO has defined five moments for hand hygiene, in this study, we monitored nine opportunities for HH. To tailor the opportunities for HH to the NICU, we monitored HH at entry and exit from the NICU main unit as well. Furthermore, to tailor the HH monitoring for our unit, where non-sterile glove use is high, we monitored HH before and after removing gloves during patient care apart from HH monitoring before and after touching patient without gloves.

The baseline overall HH compliance in our unit was low at 31.56%. The HH compliance was relatively low after touching patient care surroundings, at entry and exit from the NICU main unit, before wearing gloves and after removing gloves, at baseline and throughout the three PDSA cycles. These particular moments may be less conscious opportunities for practicing HH, leading to potential oversight. HCWs might easily miss the HH opportunities associated with touching patient care surroundings or when entering or exiting the unit since they are not in direct contact with the patients. Similarly, the compliance before and after removing gloves was also low, which was possibly due to a false sense of security among HCWs despite education [11], as demonstrated by Baloh et al. [12]. Both patients and HCWs overestimated the protective role of gloves in a study by Walaszek et al. [13]. Misuse of non-sterile gloves was identified as one of the most common causes of lower HH compliance in studies by Kurtz et al. [14] and Boudjema et al. [15].

Interestingly, the compliance with hand hygiene (HH) was observed to be higher after removing gloves compared to before wearing gloves both at baseline and throughout the PDSA cycles. Similarly, the HH compliance after touching patients was higher than before touching patients during baseline, PDSA 1, and PDSA 3. Notably, the compliance before clean or aseptic procedures and after exposure to body fluids remained consistently high, ranging from 80% to 100% throughout the study period. These findings align with a previous study conducted by Graveto et al. in 2018 [16,17], highlighting the importance of HCWs being more mindful of their personal safety.

Our Quality Improvement project successfully improved the HH compliance in our NICU unit by 15% over a 6-month period. The interventions implemented in our QI initiative included HCW education on HH, repetition of HCW education on HH and giving immediate feedback to HCWs on HH. Similar interventions have been implemented in other QI projects conducted to improve HH in various healthcare settings.

Providing immediate feedback to the HCWs on HH played a crucial role in improving HH compliance within our unit. This can be attributed to the fact that immediate feedback served as a reinforcement of their knowledge regarding various HH opportunities. It is worth noting that the median self-reported compliance of healthcare staff was consistently above 80% during individual surveys, suggesting that there were still some subconscious HH opportunities that could be effectively reinforced through immediate feedback. Additionally, the NICU head nurses who consistently demonstrated proper HH practices while giving immediate feedback may have served as role models for other HCWs, influencing their compliance positively. Furthermore, the continuous delivery of immediate feedback by the NICU head nurses throughout the day likely created a constant Hawthorne effect, promoting heightened awareness and compliance among HCWs.

The positive effect of giving immediate feedback was demonstrated in other QI projects such as that by Walker et al. in 2014 [18]. However, the immediate feedback in the study by Walker et al. was in the form of real-time dissemination of data to leadership. A multicenter study by Lehotsky et al. [19] showed improvement in hand hygiene technique with immediate personal feedback. However, a study conducted by Livshiz-Riven et al. in 2022 [20] showed that providing immediate verbal feedback did not result in a significant improvement in hand hygiene compliance during overt observation sessions. This could be because the presence of observers during the study doing overt observations might have influenced healthcare providers to adhere to proper hand hygiene practices. The addition of immediate feedback did not add to that already existing Hawthorne effect on the HCWs.

Our first two interventions, which included HCW education, failed to have a significantly positive effect on HH compliance in our unit. While several studies have demonstrated significant improvements in HH compliance through educational interventions [21,22,23,24,25], there are also studies that have shown poor compliance despite adequate HH education [26]. Furthermore, a systematic review on interventions to improve HH [27] has revealed variable outcomes following HH education. In our study, individual surveys assessing HH knowledge among HCWs indicated that 90–100% of participants were aware of all the HH opportunities, except for the HH opportunity after touching patient care surroundings, where awareness was lower at 70%. Therefore, it appears that the low HH compliance in our unit stemmed from factors other than HH knowledge, which likely contributed to the ineffectiveness of HCW education in improving compliance.

Of note, despite providing education on hand hygiene (HH) practices in healthcare settings, there were still some beliefs among healthcare workers (HCWs) that remained resistant to change. For instance, although the CDC recommends the use of alcohol-based hand sanitizers for HH due to their accessibility, ease of use, and faster application, a few HCWs continued to hold onto the belief that soap and water was the recommended method. Despite presenting them with evidence from the CDC, their adherence to this belief persisted. Similarly, HCWs also maintained a false sense of security with gloves, despite receiving evidence-based education.

We implemented repetition of HCW education as the second intervention with the aim of reinforcing hand hygiene (HH) guidelines and acting as a reminder for HCWs. However, we did not observe any significant improvement in HH rates among HCWs with repetition of HH education. This lack of improvement could be attributed to similar reasons as to why HH education failed initially. Moreover, the drop in overall HH compliance following the second intervention might be linked to an elevated patient census during that period, resulting in an increased workload for HCWs. This highlights the significance of acquiring data on workload metrics, as it would contribute to a more thorough assessment of HH compliance among HCWs. Notably, the repetition of HCW education on HH as an intervention to improve HH compliance has not been widely utilized in other quality improvement (QI) projects to the best of our knowledge.

Additional interventions that can be considered include providing individual rewards and recognition to HCWs who are compliant with hand hygiene, as suggested by the CDC [28]. Moreover, we observed that HCWs actively engaged in the project displayed greater mindfulness of HH opportunities and achieved higher levels of compliance, approaching 100%. Therefore, involving the NICU healthcare staff in the shared goal of increasing HH adherence by 20%, encouraging them to observe and provide immediate feedback to one another, could yield positive outcomes.

Strengths: Our QI project stands out due to its unique approach in monitoring HH opportunities beyond the five moments defined by the WHO. We specifically tailored the HH opportunities monitored to align with the practices in our unit. Another distinctive aspect of our project was the individualized approach taken during HCW education on HH, where each HCW received personalized attention rather than group presentations. This allowed us to identify and address various beliefs and deeply rooted misconceptions among HCWs, which can serve as valuable targets for future studies and interventions. Furthermore, our project benefited from a multidisciplinary team working collaboratively, bringing together diverse expertise and perspectives to drive meaningful improvements in HH compliance.

Limitations: In this study, the total number of observations at baseline and in all the PDSA cycles was more than 200, which provided adequate data to analyze overall HH compliance differences. However, it is important to note that the number of observations for individual HH opportunities was insufficient to accurately assess HH compliance rates. Specifically, the number of observations before clean/aseptic procedures and after exposure to patient’s body fluids was limited due to the prevalent use of gloves by HCWs in these situations.

In our study, establishing a minimum number of observations for each opportunity within a PDSA cycle proved challenging due to the nature of the observations conducted. These observations were carried out at various random times and involved diverse groups of healthcare workers (HCWs). We propose a better approach for future studies on hand hygiene compliance. The approach is to establish a predefined minimum number of observations for each opportunity category while still allowing for observations to be conducted at random times and involving various groups of HCWs. By continuing observations until this minimum threshold is met for each category, researchers can ensure a more balanced representation of hand hygiene practices across diverse situations.

Furthermore, the distribution of HH opportunities varied among different HCW roles with nurses having a greater number of opportunities compared to HCWs in other roles. Consequently, the observations were unequally distributed across various HCW roles, which should be considered when interpreting the results.

In this study, our primary focus was on HH compliance among healthcare workers (HCWs). However, we acknowledge that we did not specifically monitor whether HCWs were performing HH appropriately, following all the recommended steps for hand rub. In future QI projects aimed at improving HH compliance, it is essential to include monitoring of proper HH technique to ensure adherence to the established guideline.

Additionally, we recognize that HH compliance can be influenced by the HCWs’ work burden at any given time. Therefore, it would be beneficial to parallelly collect data on patient-to-nurse ratio or workload metrics. Understanding the correlation between HH compliance and work burden can provide valuable insights into the factors affecting HH practices and help design targeted interventions to address any challenges related to workload and compliance.

## 5. Conclusions

In conclusion, HCW education alone did not effectively improve hand hygiene (HH) compliance in our unit. However, the involvement of NICU leadership in the project and the implementation of immediate feedback yielded positive results. To further enhance HH compliance, it is important to address the false sense of security associated with glove usage among HCWs and emphasize that hand sanitizer is non-inferior to soap and water for HH in healthcare settings. Other interventions that can be considered include giving individual rewards to HCWs and involving HCWs in the shared goal of improving HH compliance. In the future, QI projects aimed at improving HH compliance among HCWs should also collect data on the technique of HH and workload metrics to better assess the HH compliance and the effectiveness of various interventions in improving the HH compliance.

## Figures and Tables

**Figure 1 children-10-01484-f001:**
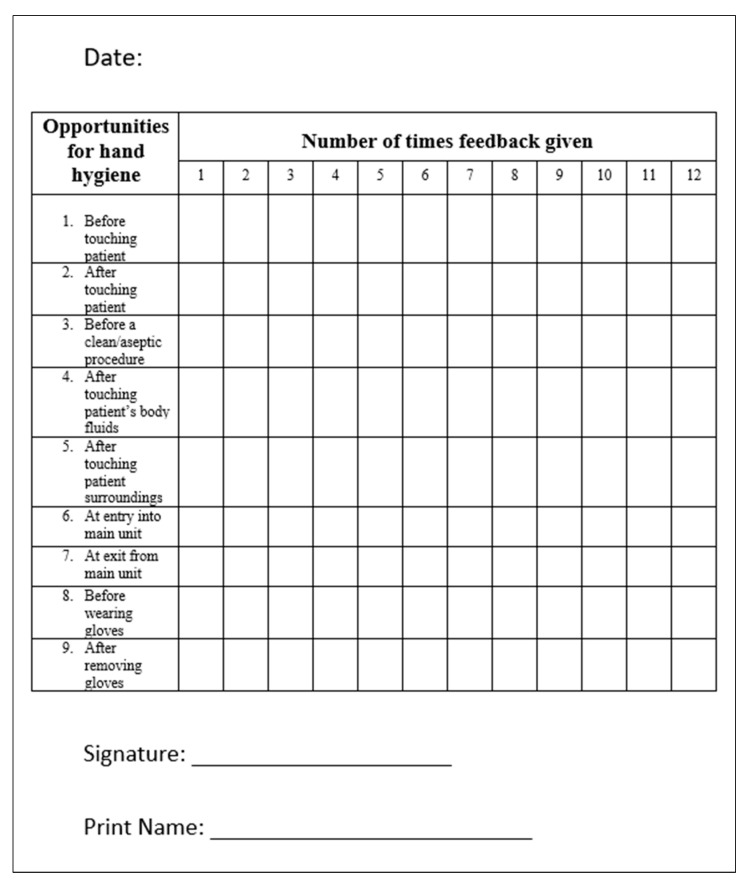
Hand hygiene immediate feedback form.

**Figure 2 children-10-01484-f002:**
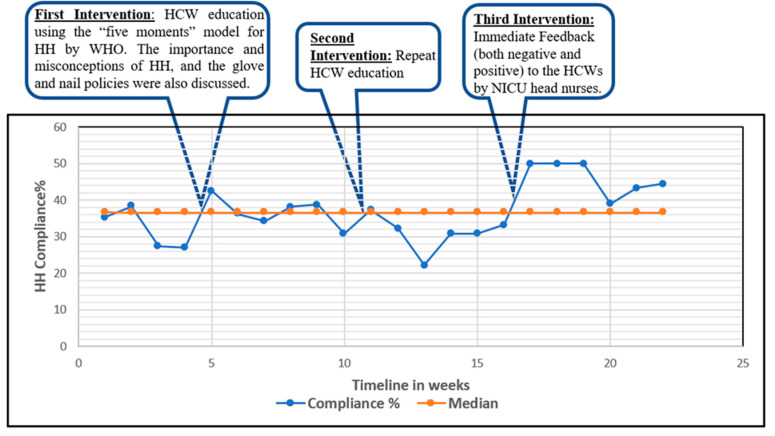
Run chart with compliance data plotted each week.

**Table 1 children-10-01484-t001:** Percentage of HH compliance at different HH opportunities and overall HH compliance at the end of each PDSA cycle.

Opportunities for Hand Hygiene	Baseline Compliance	Compliance after 1st PDSA Cycle	Compliance after 2nd PDSA Cycle	Compliance after 3rd PDSA Cycle
1—Before touching patient without gloves	36/71 (50.7%)	7/30 (56.66%)	11/19 (57.89%)	21/29 (72.4%)
2—After touching patient without gloves	18/33 (54.54%)	22/33 (66.66%)	13/25 (52.0%)	21/27 (77.77%)
3—Before a clean/aseptic procedure	9/10 (90%)	5/5 (100%)	N/A	N/A
4—After touching patient’s body fluids	8/10 (80%)	4/4 (100%)	2/3 (66.67%)	3/3 (100%)
5—After touching patient surroundings	11/62 (17.74%)	15/61 (24.59%)	5/41 (12.19%)	10/27 (37.03%)
6—At entry into main unit	15/71 (21.12%)	19/59 (32.2%)	15/49 (30.61%)	38/79 (48.10%)
7—At exit from main unit	3/47 (6.38%)	10/45 (22.22%)	6/41 (14.63%)	13/62 (20.96%)
8—Before wearing gloves	7/28 (25%)	6/26 (23.07%)	10/22 (45.45%)	11/28 (39.28%)
9—After removing gloves	12/45 (26.66%)	22/56 (39.28%)	19/39 (48.71%)	22/43 (51.16%)
Overall compliance	119/377 (31.56%)	120/319 (37.61%)	81/239 (33.89%)	139/298 (46.64%)
*p*-value *	-	0.09	0.54	<0.01

*: *p*-value for difference in HH compliance between baseline overall compliance and each PDSA cycle.

**Table 2 children-10-01484-t002:** Percentage of HH compliance at different HH opportunities and overall HH compliance at baseline and at the end of 3rd PDSA cycle.

Opportunities for Hand Hygiene	Baseline Compliance	Compliance after 3rd PDSA Cycle	*p*-Value
1—Before touching patient without gloves	36/71 (50.7%)	21/29 (72.4%)	**0.04**
2—After touching patient without gloves	18/33 (54.54%)	21/27 (77.77%)	0.06
3—Before a clean/aseptic procedure	9/10 (90%)	N/A	N/A
4—After touching patient’s body fluids	8/10 (80%)	3/3 (100%)	0.42
5—After touching patient surroundings	11/62 (17.74%)	10/27 (37.03%)	0.05
6—At entry into main unit	15/71 (21.12%)	38/79 (48.10%)	**<0.01**
7—At exit from main unit	3/47 (6.38%)	13/62 (20.96%)	**0.03**
8—Before wearing gloves	7/28 (25%)	11/28 (39.28%)	0.25
9—After removing gloves	12/45 (26.66%)	22/43 (51.16%)	**0.02**
Overall compliance	119/377 (31.56%)	139/298 (46.64%)	**<0.01**

Statistically significant *p*-values are bolded.

**Table 3 children-10-01484-t003:** Percentage of overall HH compliance among various roles of HCWs at the end of each PDSA cycle.

Role of HCW	Baseline Compliance	Compliance after 1st PDSA Cycle	Compliance after 2nd PDSA Cycle	Compliance after 3rd PDSA Cycle	*p*-Value
Attending MD	17/34 (50%)	12/21 (57.1%)	3/6 (50%)	7/13 (53.8%)	0.96
1st Year Resident	15/40 (37.5%)	8/21 (38%)	4/10 (40%)	24/34 (70.5%)	**0.02**
2nd Year Resident	14/46 (30.4%)	11/33 (33.3%)	8/16 (50%)	10/16 (62.5%)	0.09
Neonatal Fellow	10/39 (25.6%)	6/16 (37.5%)	3/7 (42.8%)	8/20 (40%)	0.6
RN	63/218 (28.8%)	83/228 (36.4%)	63/200 (31.5%)	90/215 (41.8%)	**0.03**
*p*-value	0.11	0.43	0.48	0.016	

Statistically significant *p*-values are bolded.

**Table 4 children-10-01484-t004:** Percentage of overall HH compliance among various roles of HCWs at baseline and at the end of third PDSA cycle.

Role of HCW	Baseline Compliance	Compliance after 3rd PDSA Cycle	*p*-Value
Attending MD	17/34 (50%)	7/13 (53.8%)	0.82
1st Year Resident	15/40 (37.5%)	24/34 (70.5%)	**<0.01**
2nd Year Resident	14/46 (30.4%)	10/16 (62.5%)	**0.02**
Neonatal Fellow	10/39 (25.6%)	8/20 (40%)	0.26
RN	63/218 (28.8%)	90/215 (41.8%)	**<0.01**

Statistically significant *p*-values are bolded.

## Data Availability

Data are available upon reasonable request. The data that support the findings of this study are available on request from the corresponding author. The data are not publicly available due to privacy or ethical restrictions.

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
