# Peer review of "Quality Improvement Project to Improve Hand Hygiene Compliance in a Level III Neonatal Intensive Care Unit"

_children, 2023, doi:10.3390/children10091484_

Round 1
Reviewer 1 Report
The authors submitted an interesting manuscript on an important topic for Neonatology. Hand hygiene is one of the most cost-effective interventions in NICUs world-wide and QI projects to improve hand hygiene compliance are always welcome. However, there are a few points I wish to make on this one and I will list them chronologically, both major and minor:
- Line 86 – please, be more precise of what “patient surroundings” mean (cot, incubator, radiant warmer, linens, other)
- Table 1 – the authors established a minimum number of observations for each PDSA cycle, but not a minimum number of observations for each of the nine opportunities, which lead to a significant discrepancy between the same opportunity in different PDSA cycles or between different opportunities. Also, there is no statistical analysis to tell the reader if there is statistical significance for the differences between each PDSA cycle and each opportunity;
- Table 2 – the abbreviations should be listed underneath (for example I don’t know what PGY stands for, and why PGY-1 and PGY-2 residents are accounted for separately). Also, among the different types of HCW, there is no statistical analysis to evaluate the differences between HCW roles. Also, there are no RT accounted for in this table;
- Regarding the results from both tables, there is a concerning descending trend of compliance between the second and third PDSA cycle, meaning that repeating the same teaching point as for the first cycle is useless and even detrimental to the intervention. This aspect should be further discussed in the Discussions section;
- In figure 2, please replace the dates, as they can be a bit confusing for non-American readers, with time stamps related to the project (e.g. W1, W5.... or M1, M2....);
- From what I understood in the manuscript, nail policies were only discussed, but not observed or enforced throughout the project (?);
- The Conclusion, which for the sake of continuity should be numbered 5, is way too long. The Strengths and Weaknesses subsections should be transferred to the Discussions section.
Reviewer 2 Report
Hand hygiene is very important not only in Neonatology but in every medical department.
As a neonatologist, I can confirm that we are focused on hand hygiene and disinfection. The rate of HAIs must be maintained as low as possible in the NICU because such an acquired infection can be lethal for the preemies.
The article is well written and it shows the methods used for improving HH but the article must bring some novelties to the field. I would suggest trying to add some more specific data from your NICU.
Also, you can provide info about the rate of HAIs in your NICU before and after the implementation of your interventions.
Engish is fine....minor editing required
Round 2
Reviewer 1 Report
I agree with the publication of the manuscript in its present form, I believe the authors willingness to incorporate the reviewers observations has considerably improved the manuscript.